# Aboriginal Status and Neighborhood Income Inequality Moderate the Relationship between School Absenteeism and Early Childhood Development

**DOI:** 10.3390/ijerph16081347

**Published:** 2019-04-15

**Authors:** Nazeem Muhajarine, Daphne McRae, Mohsen Soltanifar

**Affiliations:** 1Saskatchewan Population Health and Evaluation Research Unit, University of Saskatchewan, 104 Clinic Place, Saskatoon, SK S7N 2Z4, Canada; daphne.mcrae@usask.ca (D.M.); mohsen.soltanifar@mail.utoronto.ca (M.S.); 2Department of Community Health and Epidemiology, University of Saskatchewan, 107 Wiggins Rd., Saskatoon, SK S7N 5E5, Canada; 3Biostatistics Division, Dalla Lana School of Public Health, University of Toronto, 155 College St Room 620, Toronto, ON M5T 3M7, Canada

**Keywords:** school absenteeism, early childhood development, neighborhoods, Early Development Instrument, aboriginal, income inequality, multilevel modelling

## Abstract

The negative impact of school absenteeism on children’s academic performance has been documented in the educational literature, yet few studies have used validated development indicators, or investigated individual and neighborhood characteristics to illuminate potential moderating factors. Using cross-sectional Early Development Instrument (EDI) panel data (2001–2005) we constructed multilevel linear and logistic regression models to examine the association between school absenteeism and early childhood development, moderated by Aboriginal status, length of school absence, neighborhood-level income inequality, and children’s sex assigned at birth. Our study included 3572 children aged four to eight in 56 residential neighborhoods in Saskatoon, Canada. Results indicated that Aboriginal children missing an average number of school days (3.63 days) had significantly lower EDI scores compared to non-Aboriginal children, controlling for individual and neighborhood factors. As school absenteeism lengthened, the gap in EDI scores between Aboriginal and non-Aboriginal children narrowed, becoming non-significant for absences greater than two weeks. Children with long-term school absence (>4 weeks of school), living in neighborhoods of low income inequality, had significantly better physical and social development scores compared to children from medium or high income inequality neighborhoods. Across all EDI domains, girls living in neighborhoods with low income inequality had significantly better EDI scores than boys in similar neighborhoods; however, sex-differences in EDI scores were not apparent for children residing in high income inequality neighborhoods. Results add to the literature by demonstrating differences in the relationship between school absenteeism and early developmental outcomes moderated by Aboriginal status, length of school absence, neighborhood income inequality, and sex assigned at birth. These moderating factors show that differential approaches are necessary when implementing policies and programs aimed at improving school attendance.

## 1. Introduction

Over the last two decades, the negative effect of school absenteeism on children’s developmental outcomes has attracted attention in the educational and public health literature [1,2,3,4,5]. In a 2010 study from the USA [1], which included 13,613 children, school absenteeism was shown to correlate with poorer academic outcomes for kindergarten and first grade students (e.g., there was a 1.26% decrease in first grade math scores for each one standard deviation increase in school absence, *p <* 0.05). In another study including 920 fourth grade students in the USA [2], each day students were absent from school was shown to correlate with a 0.087 decrease in Palmetto Achievement Challenge Test scores (*p <* 0.0001). Aside from academic achievement, evidence suggests an association between school absenteeism and poorer childhood mental health. In a Japanese study [4], children aged seven to 17 refusing to attend school had higher mean Child Depression Inventory scores compared to controls (mean overall CDI score: 27.0 vs. 16.9; *p <* 0.05).

In Canada, there is a gap in rates of school attendance and educational outcomes at various grade levels for Aboriginal vs. non-Aboriginal students [6,7,8]. Spending less time in school (attendance rates are 10% lower for Aboriginal vs. non-Aboriginal Canadian youth [6]) likely contributes to lower rates of Aboriginal high school and post-secondary completion, and in turn, higher rates of unemployment and low adult socioeconomic status (SES) [9,10]. This line of reasoning needs to be contextualized in multigenerational and historical trauma, loss of language and cultural violence experienced by Canada’s Aboriginal peoples, especially children in the residential school system [11,12]. Against this backdrop, refining our understanding of the association between school absence and developmental well-being is particularly relevant for the current generation of Aboriginal children. By examining Aboriginal status as a moderating factor, we can determine if developmental trajectories diverge within the first year of school and if so, under what circumstances (e.g., by differing durations of school absence, neighborhood socioeconomic in/equality), to inform targeted strategies to close the Aboriginal/non-Aboriginal educational gap. As noted by Canada’s provincial Educational Ministers, there is currently a lack of evidence concerning “the stage at which Aboriginal students tend to fall behind” (early or late), and baseline measures to determine whether gaps in achievement are growing or shrinking as students age [13] (p. 11).

Studies have also shown an association between neighborhood SES and children’s developmental health (including cognitive, physical, social and emotional health) [14]. Not all studies, however, have demonstrated this relationship; further, the significance and direction of association appears to vary between studies [14]. In addition, studies examining the *compounding effect* of neighborhood material conditions and individual characteristics on children’s developmental health have reported inconsistent results [15]. To further clarify the relationship between school absenteeism and children’s development, we linked Early Developmental Instrument (EDI) data—collected from kindergarten children in Saskatoon, Canada during 2001, 2003, and 2005—with socio-demographic data from Canadian Censuses. Our objective was to model the association between school absenteeism and children’s developmental health and vulnerability, and to test whether Aboriginal status, length of school absence, neighborhood income inequality, and children’s sex assigned at birth (or combinations of these factors) modified the relationship, further exacerbating or buffering the primary relationship.

## 2. Materials and Methods

### 2.1. Data Sources and Variables

The EDI is a teacher-administered tool used to assess school readiness at the population-level [16]. It consists of 104 core questions in five general domains administered in either English or French. It is completed by the classroom teacher or early childhood educator, usually for children aged four to six, during the second half of the kindergarten year (the first year of schooling). Children receive an EDI score ranging between 0 and 10 for each of the following domains: physical health and well-being (PHWB), social competence (SC), emotional maturity (EM), language and cognitive development (L & CD), and communication and general knowledge (C & GK). Children who score in the lowest 10th percentile compared to the national level, in any one domain, are considered vulnerable [16]. In addition to developmental scores, the EDI contains individual-level demographic information such as sex, date of birth, Aboriginal status, mother tongue, neighborhood residence, and some school-based designations such as presence of special skills (i.e., numeracy, literacy) and special problems (i.e., physical disability, learning disability, behavior problem, and problems at home) as well as school absenteeism (in days) (Table 1).

From the Canadian Censuses (2001, 2006) we obtained reliable, detailed information on socioeconomic and demographic neighborhood characteristics, including annual median income, rate of employment among adults, average housing value, and income distribution [17,18]. EDI data were linked to census data using corresponding neighborhood-level geographical identifiers. Neighborhoods in Saskatoon have been defined according to long-term plans developed by the City, which reflect well-established boundaries, considered meaningful for residents. During the study period (2001 to 2006) Saskatoon had a total of 71 neighborhoods, of which 56 were considered residential with the remainder being industrial, commercial, or institutional lands.

The 2005 EDI data were linked to the 2006 Census data, the closest year of census data available. The 2003 EDI data were linked to the average of the 2001 and 2006 Census data. For the neighborhood-level characteristics of interest, we noted no significant changes in the distributions between 2001 and 2006.

Our primary predictors included individual characteristics (age, sex, Aboriginal status, maternal language, number of days absent from school, number of special skills) and residential neighborhood attributes (Gini Index, median income, unemployment rate, percentage of college graduates, average dwelling value, single parent percentage). As in previous studies, we used the Gini Index to measure neighborhood-level relative income inequality [19,20,21]. Absolute income was measured using median neighborhood income [22,23,24]. Both relative and absolute measures of income have been shown to independently influence child outcomes [23].

The primary outcomes investigated included physical health and well-being, social competence, emotional maturity, language and cognitive development, and communication skills and general knowledge, measured as individual-level EDI scores. We also used a binary variable (Yes/No) to examine whether children were vulnerable in one or more domains. All variables, with definitions and summary statistics, are presented in Table 1.

### 2.2. Statistical Analysis

We conducted linear and logistic regression multilevel modelling to account for the hierarchical structure of the data, in which children were nested within neighborhoods. Random effects were modelled for each level, using the following general linear equations:(1)$$\mathrm{Level}\;1\;\mathrm{Equation}:\;f\;\left(E\left(h_{ij}\right)\right)=\beta_{0j}+\beta_{1j}\cdot X_{1ij}+\cdot \;\cdot \;\cdot +\beta_{9j}\cdot X_{9ij}+\beta_{10j}\cdot X_{4ij}\cdot X_{6ij},$$
(2)$$\begin{array}{c} \mathrm{Level}\;2\;\mathrm{Equation}:\;\beta_{0j}=\gamma_{00}+\gamma_{01}\cdot Z_{1j}+\cdot \;\cdot \;\cdot +\gamma_{06}\cdot Z_{6j}+\gamma_{07}\cdot Z_{7j}+u_{0j},\\ \beta_{3j}=\gamma_{30}+\gamma_{31}\cdot Z_{1j}+\gamma_{32}\cdot Z_{2j},\\ \beta_{kj}=\gamma_{k0}\;\left(k\neq 3\right), \end{array}$$

For continuous outcome variables (PHWB, SC, EM, L&C, and C&GK) *h_ij_* ~ *N*(*e_ij_*, *σ*^2^) and we have *f*(*t*) = *t*. For the binary outcome variable (“vulnerability”) *h_ij_* ~ *Bernoulli*(*π_ij_*) where *π_ij_* = *P*(*h_ij_* = 1) and we have *f*(*t*) = *logit*(*t*). Here the subscript “*j*” is the index for the neighborhoods (1 ≤ *j* ≤ 56) with *n_j_* children in the *j*-th neighborhood. The subscript “*i*” is the index for the children (1 ≤ *i* ≤ *n_j_*), and the subscript “*k*” denotes the *k*-th covariable. Moreover, for the linear model *f*(*t*) = *t*, *h_ij_* stands for one of the five EDI domain outcome variables for the *i*-th child in the *j*-th neighborhood. For the logistic model *f*(*t*) = *logit*(*t*), *h_ij_* represents a “vulnerable child" for the *i*-th child in the *j*-th neighborhood. Additionally, the covariables *βkj* (1 ≤ *k* ≤ 10) are individual characteristics and the covariables *γ*0*k* (1 ≤ *k* ≤ 7) are neighborhood characteristics. Unobserved individual factors which are not correlated with individual characteristics are shown with *e_ij_*. Unlike a linear regression model, the intercept term *β*_0*j*_ given in the level two equation above varies by each neighborhood, where *γ*_00_ is the mean value of the EDI outcome for all children in Saskatoon and *u*_0*j*_ is a random quantity with normal distribution, a mean of 0, and a variance of *σ*^2^ for all children of the *j*-th neighborhood (1 ≤ *j* ≤ 56). Both of the above multilevel models were built by a chunk-wise selection method which included three chunks: chunk one (seven child variables), chunk two (seven neighborhood variables), and chunk three (three interaction variables) [25]. All models were estimated using SAS software, version 9.4 (SAS Institute Inc., Cary, NC, USA) [26]. 

## 3. Results

### 3.1. Descriptive and Inferential Statistics

Characteristics of the population sample are presented in Table 1. Of the EDI domains, physical health and well-being had the highest mean score (8.68) and communication and general knowledge had the lowest mean score (7.70). Children were on average 5.65 years old, and school absenteeism averaged 3.63 days. There was almost an equal percentage of female and male participants.

Figure 1a shows that average EDI scores for non-Aboriginal children were significantly higher than that of Aboriginal children for all EDI domains. The largest gap in scores between non-Aboriginal and Aboriginal children was evident in language and cognitive development (+1.82 units) and the smallest gap occurred in emotional maturity (+0.80 units).

Figure 1b shows a clear quadratic decreasing-increasing trend between neighborhood-level income inequality and EDI domains, especially evident for social competence and emotional maturity. Controlling for neighborhood income inequality, average EDI scores were highest in the physical health and well-being domain, while scores were the lowest in the communication and general knowledge domain. 

### 3.2. Multilevel Determinants of Developmental Outcomes

Table 2 presents the results of the final two-level linear and logistic models, indicating statistically significant associations between school absenteeism and each developmental domain, as well as overall vulnerability. All individual-level characteristics are associated with children’s scores in at least one EDI domain, when controlling for other individual and neighborhood factors and year of data collection.

Across all domains, girls, older children, and those who had special skills, had higher EDI scores compared to children without these attributes (scores were 0.437–0.933 units higher for girls, 0.212–0.624 units higher/year older, and 0.172–0.506 units higher/additional special skill). Attendance at a French immersion school was also associated with higher communication and general knowledge outcomes (0.247 units higher compared to children not attending French immersion).

At a neighborhood-level, characteristics significantly associated with better EDI scores included the percent of college educated residents, average dwelling value, and percent of single parents. For example, for each 10% increase in college educated neighborhood residents, there was a 0.019–0.023 unit increase in children’s social competence or language and cognitive development. Likewise, for each additional $10,000 in neighborhood average dwelling value, there was a 0.035–0.062 increase in children’s physical, social, and communication and general knowledge scores (see Table 2). Neighborhood unemployment rates were negatively associated with children’s development, with every 10% increase correlating with a significant decrease in physical, social, and language and communication scores (0.022–0.033 unit decrease). Median neighborhood income was not significantly associated with any of the five EDI outcomes. Year of data collection, however, was significantly associated with lower physical health and well-being scores (0.103 unit decrease/year), and increased communication and general knowledge scores (0.143 unit increase/year).

By modelling interactions, we were able to detect variation in the relationship between duration of school absence and EDI scores, depending on Aboriginal/non-Aboriginal status (Figure 2a–e). Aboriginal children with no or average school absence (3.63 days) had poorer EDI scores than non-Aboriginal children (EDI scores were 0.459–0.942 units lower for Aboriginal versus non-Aboriginal children missing an average number of school days). Statistically significant differences remained apparent for children missing up to two weeks of school; for each week of school absence, Aboriginal children’s EDI scores decreased by 0.238–0.287 units compared to non-Aboriginal children. However, as illustrated in Figure 2a–e, school absences greater than two weeks were equally associated with negative developmental outcomes for both Aboriginal and non-Aboriginal children.

Examining the moderating role of neighborhood income inequality-, we found that children with long-term (>4 weeks) school absence, residing in low income inequality neighborhoods had significantly better physical and social development scores compared to children from medium and/or high income inequality neighborhoods (Figure 3a,b). 

Assessing the moderating role of income inequality and children’s sex, we found that girls living in low income inequality neighborhoods had significantly higher EDI scores, in every domain, compared to boys in similar neighborhoods (Figure 4a–e).

However, compared to girls residing in low income inequality neighborhoods, girls from high income inequality neighborhoods had significantly lower EDI scores (0.116 units lower in physical health and well-being; 0.211 units lower in social competence; 0.356 units lower in language and cognitive development; and 0.418 units lower in communication and general knowledge) (Figure 4a,b,d,e). Neighborhood income inequality appeared to be more detrimental to girl’s development than to boy’s, as girl’s EDI scores decreased to a greater extent as income inequality increased, narrowing the sex-specific gap in EDI scores.

Pearson *χ*^2^/*df* statistics indicated good model fit for physical health and well-being (close to 1.000); moderate fit for social competence and emotional maturity (ranging from 2.000–4.000) and weak fit for communication and general knowledge (more than 4.000). However, we report harmonized results across all EDI domains, reporting the same set of variables for all outcomes.

### 3.3. Multilevel Determinants of Vulnerability

Table 2 presents the results of the logistic regression model showing the association between school absenteeism and the probability of a child being vulnerable (as defined in Section 2.1) as well as the factors that significantly moderate this association. Figure 5 shows the association between duration of school absence and the predicted probability of a child being vulnerable, moderated by Aboriginal status. 

Similar to the linear regression results, findings from the logistic model indicate an association between school absenteeism and the probability of a child being vulnerable (Table 2). Aboriginal children missing an average number of school days (3.63 days) had 2.477 times higher odds of being vulnerable compared to non-Aboriginal children (95% CI: 1.929, 3.184). However, after two weeks of school absence the gap between the two groups became negligible (Figure 5). These results are consistent with the findings from the linear model presented in Figure 2, in which Aboriginal status modified the relationship between absenteeism and EDI scores, but only for children missing up to two weeks of school.

For older children and those with a higher number of special skills, the likelihood of vulnerability decreased by 33% and 48%, respectively (aOR for one additional year of age: 0.667, 95% CI: 0.550, 0.809; aOR for each additional special skill: 0.522, 95% CI: 0.466, 0.584). In the logistic regression model, none of the neighborhood-level characteristics, nor year of data collection, significantly moderated the relationship between school absence and children being vulnerable.

## 4. Discussion

This study adds to the literature by demonstrating an association between school absence and child development, moderated by Aboriginal status, duration of absence, neighborhood income inequality, and sex assigned at birth. Among children with no school absences, Aboriginal children had lower EDI scores across all domains, compared to non-Aboriginal children, controlling for individual and neighborhood characteristics. Other Canadian studies comparing prevalence of early developmental vulnerability (measured with the EDI) between Aboriginal and non-Aboriginal children have shown both statistically significant [27], and non-significant results [28]. Lower school performance among Aboriginal students has been associated with the residual effects of colonialism (including the legacy of the residential school system) [29], greater prevalence of deprivation due to socioeconomic status [30,31], and lower funding for on-reserve compared to off-reserve, public school [32]. In addition, lower Aboriginal academic achievement has been linked to experiences of racism and marginalization, including low teacher expectations based on biased social beliefs [33] and disruptions in education because of frequent transition between schools and school districts [34].

Results showed no difference, however, in EDI scores or vulnerability for Aboriginal versus non-Aboriginal children missing more than two weeks of school. These findings suggest, at a population-level, non-Aboriginal children with longer-term absenteeism (>2 weeks) may be facing multiple, systematic barriers to early development, much like their Aboriginal peers with longer absences. For children missing more than two weeks of school (Aboriginal or non-Aboriginal), median neighborhood household income was $6800 less than students missing two weeks of school or less. Likewise, children with longer-term absenteeism resided in neighborhoods with higher rates of single-parent families (18.34% vs. 13.15%) and unemployment (10.21% vs. 6.32%), and in neighborhoods with fewer college-educated adults (11.20% vs. 19.80%) compared to children with short term (<2 weeks) school absence. Neighborhood-level disadvantage may contribute to school absence due to a contagion effect [35], where parents’ and children’s behaviors and attitudes toward school attendance mirror their neighbors’ over time. Moreover, the lack of material and social resources in neighborhoods with low SES may make it difficult for children to attend school on a consistent basis. For example, in lower SES neighborhoods fewer private vehicles and lower levels of trust between neighbors may diminish options for transportation (e.g., car-pooling), in turn impacting school attendance.

Early child development appeared to be compromised by short-term school absence (<2 weeks) for Aboriginal children, and a greater average number of days missed (7.64 days on average) compared to non-Aboriginal children (2.80 days on average). Although longer absenteeism (>2 weeks) was associated with greater likelihood of child developmental vulnerability for Aboriginal and non-Aboriginal children, it was 12.4 times more likely among Aboriginal children. Results suggest that differences in both prevalence of school absence and associations between school absence and outcomes may, at a population-level, contribute to greater child developmental vulnerability for Aboriginal versus non-Aboriginal children. 

These findings indicate the need for greater understanding of the factors contributing to Aboriginal children’s school absence, in order to develop effective strategies to improve attendance. An understanding of the mediating factors leading to higher than average absenteeism, and the greater impact of short-term school absence on Aboriginal children’s early development requires input from Aboriginal stakeholders who can prove culturally relevant perspectives. Future research exploring barriers and facilitators to school attendance should be guided, interpreted, and integrated into a community response by the Aboriginal communities, parents and children directly affected.

High neighborhood-level income inequality also appeared to negatively impact children’s EDI scores, particularly amongst girls. Competition and relative deprivation theories may help to explain these outcomes [14]. According to competition theories, in high income inequality neighborhoods individuals of lower socioeconomic status (SES) competing for scarce resources (e.g., school or employment opportunities) will usually “lose” to those of higher SES, generating or reinforcing feelings of inadequacy and social exclusion [35,36]. Applying the principles of relative deprivation theory, parents and children of lower SES who live in neighborhoods with high income inequality (a mix of low and high SES families) may frequently compare themselves to their higher SES peers, inducing a sense of inferiority and low self-esteem that can affect developmental well-being [35,36]. Conversely, children who reside in low income inequality neighborhoods may be shielded from the stress of status-related comparisons, assuming their neighbors have relatively similar material resources. Young children of low SES may benefit from living in low income inequality neighborhoods as they may not be fully aware of their social position in larger society, therefore comparisons may be limited to those in close proximity, such as neighborhood peers. 

### Limitations

Some researchers have questioned the validity of the EDI for assessing Aboriginal children’s school readiness as their cultural, language, and lived experience may meaningful differ from that of the teachers administering the instrument, and the EDI may not capture culturally valued behaviors, skills, and knowledge [37]. However, studies specifically investigating the validity of the EDI as an assessment tool for Aboriginal children have demonstrated no implementation bias due to the administration of the EDI by teachers [38], nor bias in the EDI instrument itself [39,40]. Nonetheless, the interpretation of results requires active and continuous engagement with local Aboriginal communities to ensure their knowledge informs the meaning of the study findings and the next steps in transforming results into culturally appropriate and tailored action [38].

## 5. Conclusions

Research has demonstrated that investment in early childhood policies and effective intervention programs for vulnerable children increase quality of life and offer the greatest financial returns to society [41,42]. This study provides evidence of the critical role of regular school attendance for kindergarten children’s early development. Kindergarten school absence appears to be the most detrimental for Aboriginal children, and those living in neighborhoods of high income inequality—particularly for girls. Non-Aboriginal children with longer-term school absence (>2 weeks) are also among those at greatest risk of developmental vulnerability. As this study shows, improving school attendance has the potential to enhance early childhood development, at a population-level. Future research should include qualitative input from the communities, parents, and children studied, to understand their perceptions of obstacles to school attendance and to promote the development of locally-relevant policies and programs. 

## Figures and Tables

**Figure 1 ijerph-16-01347-f001:**
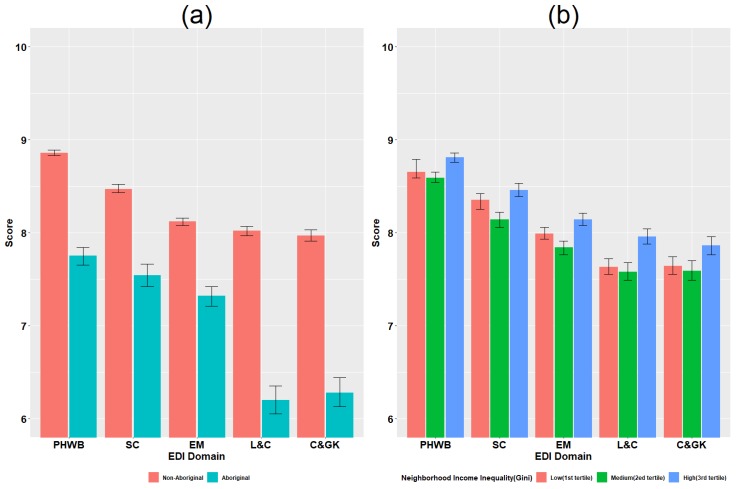
EDI scores by Aboriginal/non-Aboriginal status and neighborhood income inequality. (**a**) compares mean EDI scores and 95% confidence intervals by EDI domain for Aboriginal versus non-Aboriginal children. (**b**) compares mean EDI scores and 95% confidence intervals by EDI domain for neighborhoods of low, medium and high income inequality. Abbreviations: PHWB: physical health and well-being; SC: social competence; EM: emotional maturity; L&C: language and cognitive development; C & GK: communication and general knowledge.

**Figure 2 ijerph-16-01347-f002:**
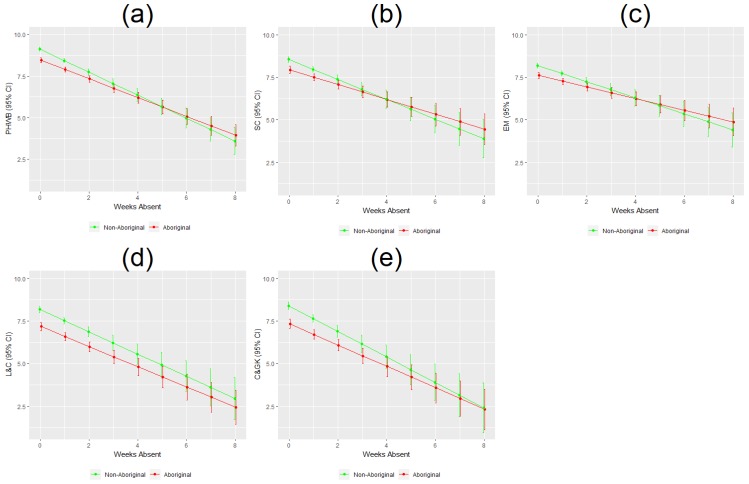
Predicted EDI scores by duration of school absence and Aboriginal/non-Aboriginal status. (**a**–**e**) show the relationship between duration of school absence in weeks and predicted EDI domain scores and 95% confidence intervals are different for Aboriginal versus non-Aboriginal children. PHWB: physical health and well-being; SC: social competence; EM: emotional maturity; L & C: language and cognitive development; C & GK: communication and general knowledge.

**Figure 3 ijerph-16-01347-f003:**
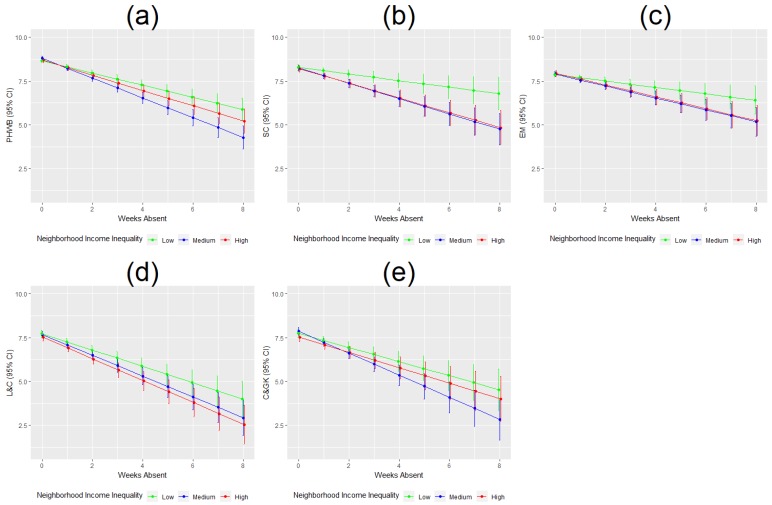
Predicted EDI scores by duration of school absence and neighborhood income inequality. (**a**–**e**) show the relationship between duration of school absence in weeks and predicted EDI domain scores and 95% confidence intervals are different for neighborhoods of low, medium and high income inequality. PHWB: physical health and well-being; SC: social competence; EM: emotional maturity; L & C: language and cognitive development; C & GK: communication and general knowledge.

**Figure 4 ijerph-16-01347-f004:**
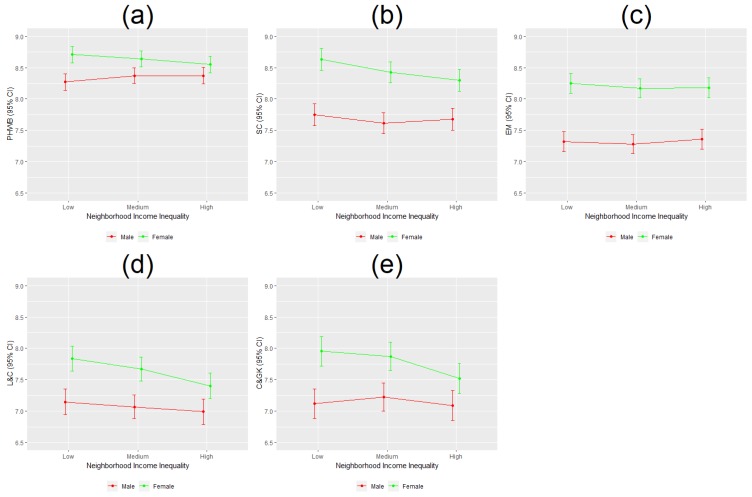
Predicted EDI scores by neighborhood income inequality and children’s sex. (**a**–**e**) show the relationship between low, medium and high neighborhood income inequality and predicted EDI domain scores and 95% confidence intervals are different for male versus female children. PHWB: physical health and well-being; SC: social competence; EM: emotional maturity; L & C: language and cognitive development; C & GK: communication and general knowledge.

**Figure 5 ijerph-16-01347-f005:**
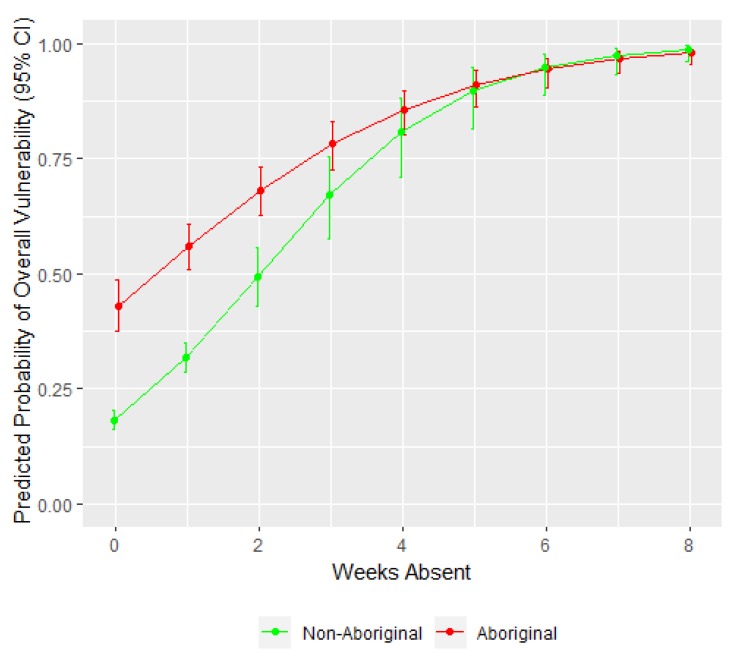
Predicted probability of a child being vulnerable by Aboriginal status and duration of school absence. The figure compares the relationship between the number of weeks children were absent from school and the predicted probability of a child being vulnerable, and 95% confidence intervals, for Aboriginal versus non-Aboriginal children.

**Table 1 ijerph-16-01347-t001:** Descriptive statistics of study variables (*n* = 3572).

Variable	Category	Mean ± se	*n*%	Min–Max
*Predictors*				
Children’s Characteristics				
Age		5.65 ± 0.0042		4.01–7.94
Days absent		3.63 ± 0.0610		0.00–56.00
Number of special skills		0.43 ± 0.0129		0.00–7.00
Sex	Female		49.18	
	Male		50.82	
Aboriginal	Yes		17.15	
	No		82.85	
Attendance at French immersion school	Yes		11.53	
	No		88.47	
Maternal language	English		93.85	
	Other		6.15	
Neighborhood Characteristics				
Gini Index		0.42 ± 0.004		0.31–0.60
Median income ($10 K/capita)		2.53 ± 0.0090		0.61–4.12
Unemployment rate (15+ years of age)		6.44 ± 0.0499		1.30–27.70
College educated percentage		19.52 ± 0.1334		1.00–65.00
Dwelling average value ($10 K)		17.49 ± 0.0765		6.81–33.23
Single parent percentage		13.32 ± 0.0772		0.00–42.59
*Continuous Outcomes (EDI)*				
Physical health and well-being		8.68 ± 0.0163		1.53–10.00
Social competence		8.32 ± 0.0216		0.00–10.00
Emotional maturity		8.00 ± 0.0193		1.00–10.00
Language & cognitive development		7.73 ± 0.0255		0.00–10.00
Communication & general knowledge		7.70 ± 0.0290		0.00–10.00
*Binary Outcome*				
Vulnerable	Yes		28.71	
	No		71.29	

**Table 2 ijerph-16-01347-t002:** Two-level generalized linear model and logistic model estimates (*n* = 3572).

	Linear Model					Logistic Model
Outcomes	PHWB	SC	EM	L & C	C & GK	Vulnerable
Predictors						
Children’s Characteristics						
Constant (±s.e)	6.152 *** (±0.3708)	5.054 *** (±0.5047)	5.899 *** (±0.4537)	4.036 *** (±0.5701)	3.592 *** (±0.6720)	2.327 *** (±0.6914)
Age (±s.e)	0.212 *** (±0.0465)	0.366 *** (±0.0653)	0.213 *** (±0.0585)	0.624 *** (±0.072)	0.397 *** (±0.0848)	−0.405 *** (±0.0985)
Days absent (±s.e)	−0.067 *** (±0.0075)	−0.047 *** (±0.0104)	−0.045 *** (±0.0094)	−0.075 *** (±0.0116)	−0.074 *** (±0.0136)	0.086 *** (±0.0155)
Number Special Skills (±s.e)	0.235 *** (±0.0152)	0.285 *** (±0.0213)	0.172 *** (±0.0192)	0.467 ** (±0.0236)	0.506 *** (±0.0278)	−0.650 *** (±0.0587)
$\mathrm{Sex}{\;}^{1}\;(\pm s.e$)	0.437 *** (±0.0536)	0.878 *** (±0.075)	0.933 *** (±0.0677)	0.691 *** (±0.083)	0.838 *** (±0.0979)	−0.934 *** (±0.1155)
$\mathrm{Aboriginal}{\;}^{2}\;(\pm s.e$)	−0.671 *** (±0.0630)	−0.608 *** (±0.0844)	−0.596 *** (±0.079)	−1.004 *** (±0.0977)	−1.063 *** (±0.1150)	1.111 *** (±0.1248)
$\mathrm{Attendance}\;\mathrm{at}\;\mathrm{French}\;\mathrm{Immersion}\;\mathrm{school}{\;}^{2}\;(\pm s.e$)	0.080 (±0.0503)	−0.092 (±0.0706)	−0.016 (±0.0637)	−0.002 (±0.0781)	0.247 ** (±0.0917)	−0.107 (±0.1087)
$\mathrm{English}\;\mathrm{as}\;\mathrm{maternal}\;\mathrm{language}{\;}^{2}\;(\pm s.e$)	0.002 (±0.0030)	0.003 (±0.0042)	−0.001 (±0.0038)	−0.004 (±0.0047)	−0.029 *** (±0.0055)	0.009 * (±0.0058)
Neighborhood Characteristics						
$\mathrm{Gini-Medium}{\;}^{3}\;(\pm s.e$)	0.213 ** (±0.0812)	−0.006 (±0.1135)	0.045 (±0.1020)	−0.013 (±0.1292)	0.225 (±0.1493)	−0.094 (±0.1441)
$\mathrm{Gini-Medium}{\;}^{3}\;(\pm s.e$)	0.144 (±0.0928)	0.045 (±0.1270)	0.125 (±0.1143)	−0.074 (±0.1458)	−0.009 (±0.1698)	−0.059 (±0.1570)
Median income (±s.e)	0.109 (±0.0908)	−0.094 (±0.1259)	0.135 (±0.1142)	−0.086 (±0.1455)	0.018 (±0.1744)	−0.207 (±0.1535)
Unemployment rate (±s.e)	−0.022 ** (±0.0092)	−0.024 * (±0.0129)	−0.010 (±0.0117)	−0.033 ** (±0.0153)	−0.014 (±0.0171)	0.021 (±0.0146)
College educated percentage (±s.e)	0.000 (±0.0050)	0.019 ** (±0.0068)	0.007 (±0.0059)	0.023 *** (±0.0076)	0.013 (±0.0091)	−0.009 (±0.0081)
Dwelling Average Value (±s.e)	0.046 ** (±0.0147)	0.035 * (±0.0168)	−0.000 (±0.0168)	0.007 (±0.0211)	0.062 ** (±0.0263)	−0.006 (±0.0229)
Single Parents Percentage (±s.e)	0.025 ** (±0.0081)	0.025 ** (±0.0107)	0.009 (±0.0096)	−0.003 (±0.0123)	0.027 * (±0.0147)	−0.007 (±0.0129)
$\mathrm{Year}{\;}^{4}\;(\pm s.e$)	−0.103 ** (±0.0313)	−0.058 (±0.0415)	0.037 (±0.0369)	0.064 (±0.0469)	0.143 ** (±0.0568)	−0.007 (±0.0494)
Interactions						
$\mathrm{Days}\;\mathrm{absent}\;*\;\mathrm{Gini-Medium}{\;}^{3}\;(\pm s.e$)	−0.030 *** (±0.0085)	−0.035 *** (±0.0119)	−0.022 ** (±0.0106)	−0.018 (±0.0132)	−0.0326 ** (±0.0155)	0.022 (±0.0177)
$\mathrm{Days}\;\mathrm{absent}\;*\;\mathrm{Gini-High}{\;}^{3}\;(\pm s.e$)	−0.012 (±0.0090)	−0.033 ** (±0.0126)	−0.022 * (±0.0113)	−0.022 (±0.0141)	−0.005 (±0.0165)	0.011 (±0.0190)
Days absent * Aboriginal (±s.e)	0.036 *** (±0.0075)	0.041 *** (±0.0106)	0.038 *** (±0.0095)	0.017 (±0.0117)	0.034 ** (±0.0137)	−0.056 *** (±0.0159)
Female * Gini-Medium (±s.e)	−0.164 ** (±0.0765)	−0.067 (±0.1076)	−0.040 (±0.0966)	−0.088 (±0.1189)	−0.192 (±0.1397)	0.057 (±0.1643)
Female * Gini-High (±s.e)	−0.260 *** (±0.0752)	−0.256 ** (±0.1057)	−0.113 (±0.0950)	−0.282 ** (±0.1168)	−0.409 *** (±0.1373)	0.115 (±0.1659)
Model Information						
$\sigma_{u}^{2}\left(Neighborhood\right)$	0.034	0.040	0.034	0.063	0.094	0.027
$\sigma_{e}^{2}\left(Residual\right)$	1.269	2.503	2.007	3.048	4.223	3.290
VPC (%)	2.6	1.5	1.7	2.0	2.1	0.8
AIC	16,729.2	20,363.1	19,061.8	21,391.5	23,167.1	5603.7
$\mathrm{Pearson}\;\chi^{2}/df$	1.322	2.582	2.075	3.168	4.393	1.030

Notes: *** Indicates *p*-value < 0.005; ** Indicates *p*-value < 0.05; * Indicates *p*-value < 0.10; ^1^ Reference category is “male”; ^2^ Reference category is “no”; ^3^ Reference category is “Gini-low”; ^4^ Normalized to −2, 0, +2; PHWB: physical health & wellbeing, SC: social competence, EM: emotional maturity, L & C: language & cognitive development, C & GK: communication & general knowledge; VPC: variance partition coefficient.

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
