# Peer review of "Aboriginal Status and Neighborhood Income Inequality Moderate the Relationship between School Absenteeism and Early Childhood Development"

_ijerph, 2019, doi:10.3390/ijerph16081347_

Round 1

Reviewer 1 Report

I feel that the Introduction should explain why Aboriginal status is included in the analyses. Part of the the authors' plan was to model the association between school absenteeism and children’s developmental health and vulnerability, and to test whether Aboriginal status (along with other variables) modified the relationship. It's important to explain why the authors chose to include this variable in the model. Background information on this potential relationship would be helpful.

I think it might be important to include a measure of vulnerability for each of the 5 domains included in the model, rather than a single, general measure of vulnerability. The 5 domains are measuring very different concepts, so it seems odd to not differentiate them in a measure of vulnerability.

Author Response

Comments from Referee 1

Authors’ Response

“I feel that the Introduction should   explain why Aboriginal status is included in the analyses.”

“It’s important to explain why the   authors chose to include this variable in the model. Background information   on this potential relationship would be helpful.”

Thank you for your comments. We have revised the manuscript to include a paragraph in the introduction explaining, in the Canadian context, differences in Aboriginal school attendance and student achievement compared to non-Aboriginal students. We have also expanded on our rationale for including this variable in the study.

“I think it might be important to include   a measure of vulnerability for each of the 5 domains included in the model,   rather than a single, general measure of vulnerability. The 5 domains are   measuring very different concepts, so it seems odd to not differentiate them   in a measure of vulnerability.”

We agree that the 5 EDI domains are measuring different specific developmental concepts, but they are related and can and have been considered as an overall measure of developmental achievement or lack of achievement (i.e. vulnerability). Therefore, for the   linear modelling we tested the association between school absenteeism and each individual EDI domain. We then re-ran our analyses using a logistic   model and a single measure for vulnerability, to confirm our initial   domain-specific results. Results were similar in direction and significance   between the two types of models for the individual characteristics and the   interaction between Aboriginal status and duration of school absence. By using two different modelling methods and two types of outcomes (domain-specific and an aggregated outcome), we feel that we have provided a more rigorous and comprehensive analysis than if we had limited our modelling to one type of method or outcome.

Reviewer 2 Report

It is a good paper — good data and a strong analysis. However, the presentation is so terse that it is hard to understand. The figures, for example, should aid understanding but they take time o figure out. Also, the authors should eschew all causal language; these are associations. The Discussion is thoughtful and it contains the motivations underlying the hypotheses; but this needs to be incorporated in the introduction. I appreciate a short paper and the research underlying the main associations is clear; but the interactions / moderating factors are not as clear. In sum, the analyses are fine; but the authors need to be more clear about why they conducted the study, how to read the results, and what the results are.

Author Response

Comments from Referee 2

Authors’ Response

“. . . the presentation is so terse that it is hard to understand. The figures, for example, should aid understanding   but they take time to figure out.”

Thank you for your comments. We have added figure legends to help clarify the presentation.

“Also, the authors should eschew all   causal language; these are associations.”

Please see the tracked changes throughout   the Abstract, Results and Discussion sections. Language implying causality   has been retracted.

“The Discussion is thoughtful and it   contains the motivations underlying the hypotheses; but this needs to be   incorporated in the introduction.”

Please see the paragraph added to the Introduction on the second page. 

“. . . the interactions/moderating factors are not as clear.”

“. . . the authors need to be more clear   about why they conducted the study, how to read the results, and what the   results are.”

In the Introduction we have expanded on the rationale for the study and included more explanation on our selection of  “Aboriginal/non-Aboriginal status” as a study variable. In response to your   comments, we have included figure legends to foster a clearer understanding   of the results.